# Slowly folding surface extension in the prototypic avian hepatitis B virus capsid governs stability

Cihan Makbul[1], Michael Nassal[2†]*, Bettina Böttcher[1†]*

[1]Julius Maximilian University of Würzburg, Department of Biochemistry and Rudolf Virchow Centre, Würzburg, Germany; [2]University Hospital Freiburg, Internal Medicine 2/Molecular Biology, Freiburg, Germany

**Abstract** Hepatitis B virus (HBV) is an important but difficult to study human pathogen. Most basics of the hepadnaviral life-cycle were unraveled using duck HBV (DHBV) as a model although DHBV has a capsid protein (CP) comprising ~260 rather than ~180 amino acids. Here we present high-resolution structures of several DHBV capsid-like particles (CLPs) determined by electron cryo-microscopy. As for HBV, DHBV CLPs consist of a dimeric α-helical frame-work with protruding spikes at the dimer interface. A fundamental new feature is a ~ 45 amino acid proline-rich extension in each monomer replacing the tip of the spikes in HBV CP. In vitro, folding of the extension takes months, implying a catalyzed process in vivo. DHBc variants lacking a folding-proficient extension produced regular CLPs in bacteria but failed to form stable nucleocapsids in hepatoma cells. We propose that the extension domain acts as a conformational switch with differential response options during viral infection.

*For correspondence:
nassal2@ukl.uni-freiburg.de (MN);
bettina.boettcher@uni-wuerzburg.de (BBö)

[†]These authors contributed equally to this work

Competing interests: The authors declare that no competing interests exist.

## Introduction

Hepadnaviruses (*hepa*totropic *DNA* viruses) are a family of small enveloped retrotranscribing DNA viruses. In humans, chronic infections with HBV cause ~800,000 deaths per year (*Report, 2017*). Current therapies can control but rarely cure infection (*Fanning et al., 2019*; *Martinez et al., 2019*; *Nassal, 2015*). Most fundamental aspects of the hepadnaviral lifecycle such as replication of the tiny ~3 kb DNA via reverse transcription of a pregenomic (pg) RNA (*Summers and Mason, 1982*), or the use of a covalently closed circular (ccc) DNA form (*Tuttleman et al., 1986*) as episomal persistence reservoir (*Nassal, 2015*), were uncovered using more tractable model viruses, mostly duck HBV (DHBV) (*Schultz et al., 2004*), prototype of the genus *avihepadnaviridae*.

Despite basal similarities in genome size, genetic organization and replication strategy the avian viruses exhibit characteristic differences to human and the other mammalian HBVs (genus *orthohepadnaviridae*). They lack an X protein, a regulator of cccDNA transcription (*Livingston et al., 2017*), and a medium sized (M) envelope or surface protein besides the small (S) and large (L) forms; in addition, the avian virus surface proteins are smaller, with only three predicted transmembrane helices versus four in the orthohepadnaviruses. Particularly striking is the much larger size (~260 amino acids) of their single capsid or core proteins (CPs) which in mammalian HBVs comprise only ~180 aa. Recently discovered HBV-like viruses in distant hosts (*Lauber et al., 2017*) maintain this division. Most employ the small type CP but amphibian and reptilian viruses, e.g. Tibetan frog HBV (*Dill et al., 2016*) and lizard HBVs (*Lauber et al., 2017*), encode the large type CP. As the small CPs can obviously perform all essential CP functions (see below, and *Venkatakrishnan and Zlotnick, 2016*) the biology behind the large type CPs is enigmatic. A key step would be knowledge of their structure but, ironically, here DHBV lags much behind HBV for which various structures are available for recombinant capsid-like particles (CLPs) (*Böttcher and Nassal, 2018*; *Böttcher et al., 1997*;

*Conway et al., 1997*; *Crowther et al., 1994*; *Wynne et al., 1999*), serum-derived nucleocapsids (*Roseman et al., 2005*) and virions (*Dryden et al., 2006*; *Seitz et al., 2007*).

CPs are multifunctional (*Zlotnick et al., 2015*; *Diab et al., 2018*), making the HBV CP (HBc) a prime novel antiviral target (*Lahlali et al., 2018*; *Schlicksup et al., 2018*). Most obvious is their role as the building block of the icosahedral viral capsid. Production of progeny virions during infection involves transcription from nuclear cccDNA, translation in the cytoplasm of CP, polymerase and the other viral proteins, CP phosphorylation-regulated encapsidation of a complex of pgRNA and polymerase (*Heger-Stevic et al., 2018b*; *Zhao et al., 2018*), and reverse transcription of pgRNA into relaxed circular (RC plus some double-stranded linear (dsL)) DNA. Such nucleocapsids can interact with the surface proteins to be secreted as enveloped virions. Alternatively, they may interact with nuclear import receptors to be redirected into the nuclear pore where they disintegrate (*Gallucci and Kann, 2017*), releasing the rcDNA for conversion into cccDNA (*Nassal, 2015*; *Schreiner and Nassal, 2017*).

How hepadnaviral CPs integrate the different interaction cues to follow one path or the other is poorly understood but must somehow relate to their three-dimensional (3D) structure. Typical small type CPs comprise an N-terminal ~140 aa assembly domain (*Birnbaum and Nassal, 1990*) and an arginine-rich ~34 aa C-terminal domain (CTD) whose nucleic acid binding capacity is modulated by phosphorylation (*Heger-Stevic et al., 2018b*; *Zhao et al., 2018*). Both domains are connected by an ~10 aa linker (*Watts et al., 2002*).

The assembly domains feature five α-helices (*Figure 1—figure supplement 1*). α3 and α4 form an antiparallel hairpin providing the interface for dimerization with a second subunit. The resulting four-helix bundles protrude as spikes from the capsid surface. The helix α5 plus the downstream sequence (the 'hand region') provide the inter-dimer contacts, mostly arranged in T = 4 icosahedral symmetry which comprises 120 Cp dimers (the minor T = 3 class encompasses 90 dimers). The asymmetric unit is composed of two dimers in spatially different surroundings with four quasi-equivalent monomer conformations, A to D (*Figure 1—figure supplement 1*). The first residues of the CTD are capsid-internal (*Zlotnick et al., 1997*) but residues past position 150 usually lack sufficient order for visualization (*Böttcher and Nassal, 2018*).

Early cryo-EM reconstructions of DHBc suggested a T = 4 arrangement similar to HBc CLPs but with laterally widened spikes (*Kenney et al., 1995*). Extensive mutagenesis led to a DHBc model (*Nassal et al., 2007*; *Vorreiter et al., 2007*) predicting an HBc-like body of dimeric assembly domains comprising the N-terminal ~180 aa, an ~80 aa CTD with a more profound morphogenetic impact than in HBc, and a much extended ~45 aa proline-rich loop (residues 78–122) connecting helices α3 and α4. However, all attempts to directly verify this model by cryo-EM reconstructions fell short of sufficient resolution. The broader distribution of DHBc vs. HBc CLPs during sedimentation in sucrose gradients and native agarose gel electrophoresis (NAGE) suggested that morphological heterogeneity obscured the data. Some DHBc variants such as R124E produced more homogeneous CLPs (termed class2 *Nassal et al., 2007*) but did not yield better resolved cryo-EM reconstructions. In particular, the predicted loop at the spike tips remained invisible, in line with complete disorder.

Here we discovered that over several months of storage this loop, now termed extension domain, adopted a defined fold. This enabled us to determine the structure of whole DHBc CLPs at a resolution of 3.7 Å enabling de novo model building. Functional studies revealed that the extension domain´s ability to fold is dispensable for recombinant CLP assembly but correlates with capsid stability in hepatoma cells. We propose that plasticity of the extension domain represents a large-type CP-specific evolutionary adaptation.

## Results

DHBV CP (DHBc) expressed in *E. coli* self-assembles into CLPs which package bacterial RNA (*Kenney et al., 1995*; *Nassal et al., 2007*). Here, improved expression was achieved using vectors (*Heger-Stevic et al., 2018b*) with an *E. coli* optimized DHBc gene encoding the DHBV16 CP sequence (Uniprot accession: P0C6J7.1). CLPs for cryo-EM analysis were enriched by two sequential ammonium sulphate precipitations (*Birnbaum and Nassal, 1990*) followed by sucrose gradient sedimentation (*Heger-Stevic et al., 2018a*; *Figure 1—figure supplement 2*).

After two weeks of storage at 4°C we exchanged the sucrose-containing buffer against a cryo-EM compatible buffer with low solute concentration and vitrified the samples. Micrographs showed

distinct particles of spherical shape, aggregates of such particles and some proteo-liposomes (*Figure 1—figure supplement 2*). 2D-class averages revealed particles with protruding spikes but also particles that lacked distinct spikes or were deformed (*Figure 1—figure supplement 2*), indicating that only a fraction of particles had assembled into regular icosahedral capsids. The resulting 3D-map had a nominal resolution of 6.1 Å and showed capsids with an arrangement of spikes and holes resembling that of HBc-capsids with T = 4 triangulation (*Crowther et al., 1994*) but with 15 Å broader and 5 Å longer spikes (*Figure 1—figure supplement 3*). The characteristic broad spikes of the DHBc CLPs were only visible at lower density thresholds but disappeared at higher density thresholds, suggesting a markedly higher structural variability in the extensions than in the core of the spikes.

To test this hypothesis, we analysed the individual asymmetric units by 3D-classification. The class averages (*Figure 1—figure supplement 3*) showed 42% of asymmetric units without proper spikes and lower average density consistent with dimers either being misfolded, largely displaced or completely missing. Another third of the asymmetric units grouped into classes in which both spikes were broad, suggesting that all extension domains in the asymmetric unit were folded. The remaining 25% of asymmetric units grouped into classes with one narrow and one broad spike, consistent with folded extension domains in the broad spikes and still unfolded or flexible ones in the narrow spikes. Furthermore, the radial position of the spikes in the CLPs varied by up to 7 Å, indicating local deformations of the capsid shell.

The analysis of the individual asymmetric units required accurate determination of the orientations of an ordered asymmetric unit of only some 80 kDa, which is at the lower size limit that can be confidently analysed. We speculated that our analysis might have been limited by the imaging conditions (integrating mode, exposure <40 e/Å$^2$), which were chosen for entire capsids (*Song et al., 2019*) rather than for the individual asymmetric units. Therefore, we acquired another data set at a higher dose in counting mode (*Table 1*). For this we prepared new grids of the CLPs that had now been stored in sucrose at 4°C for 10 months.

Much to our surprise 2D-class averages showed crisper projection averages (*Figure 1—figure supplement 2*) and the 3D-maps of the CLPs were better resolved than before. The 3D-map of the CLPs had a resolution of 3.7 Å (*Figure 1*, *Figure 1—figure supplement 4*), but more importantly, the extension domains were clearly visible at the same threshold as the core of the spikes, indicating that during long-term storage most extension domains had become folded. To investigate the folding of individual extension domains, we analysed the asymmetric units as described above. We observed three well resolved classes with broad spikes comprising 74% of the asymmetric units (*Figure 1*, *Figure 1—figure supplement 4*), one class with broad spikes and low resolution (9% of the asymmetric units), and one class with a very low average density (18% of the asymmetric units), in line with a still significant number of missing or grossly displaced asymmetric units in the capsids. The three well-resolved classes varied in their radial position by only 2 Å with no major structural differences. Local refinement of the orientations of these particles further improved the overall resolution of the asymmetric unit to a nominal resolution of 3 Å (*Figure 1—figure supplement 4*) with the extension domains still less well resolved than the rest of the asymmetric unit.

We were able to build a de novo model of DHBc's assembly domain into the 3D-map of the asymmetric unit of DHBc (*Figure 1*, *Figure 1—figure supplement 4*, *Table 2*). All four subunits of the asymmetric unit had the same overall fold with adaptations to the different spatial surroundings close to the N-termini and in the C-terminal hand regions (*Figure 2a–c*). The core of the spikes was formed by four helices, with each DHBc monomer contributing an ascending helix (α3; residues 44–78, 35 aa) and a descending helix (α4; residues 120–157, 38 aa). The ascending helix α3 was kinked in the center between F60 and W61 and tilted away from the dimer axis in the upper part of the spike. The helices were considerably longer than in HBc (*Figure 3* and *Figure 3—figure supplement 1*), where α3 is straight (25 aa) and α4 is kinked (31 aa). The four helices of the spike of HBc and DHBc had a very similar packing below the kink (*Figure 3*) but diverged above the kink with a different twist of the helices around each other (*Figure 3*). At the tips of the spikes the positions of the helices were dramatically different, basically swapping the positions of DHBc α3 with HBc α4 in the same subunit and of DHBc α4 with HBc α3 between different subunits of the dimer. Moreover, DHBc helices α3 and α4 were connected through a 41 aa long extension domain (residues 79–119, *Figure 2g*) whereas in HBc this connection is just a short loop of 3–4 aa. The extension with two short helical segments (αe1: 92–99; αe2: 103–117), was positioned at the side of the spike close to

**Table 1.** Summary of image reconstruction and model building.

| Sample prepared @ | DHBc 2 weeks | DHBc 10 months | DHBcR124EΔ 2 weeks | DHBc+FkpA 2 weeks | DHBcR124E 3 months | |
|---|---|---|---|---|---|---|
| Magnification | 75,000 | | | | | |
| Acquisition mode | integrating mode | counting mode | Counting mode | Counting mode | Integrating mode | Integrating mode |
| Total exposure | 39 e⁻/Å² | 77 e⁻/Å² | 60 e⁻/Å² | 60 e⁻/Å² | 40 e⁻/Å² | 40 e⁻/Å² |
| Exposure time | 3.5 s | 75 s | 54 s | 56.6 s | 2.4 s | 2.3 s |
| Pixel size | 1.0635 Å | | | | | |
| Beam diameter | 1 μm | | | | | |
| C2 aperture | 70 μm | | | | | |
| Objective aperture | 100 μm | | | | | |
| Spot size | 5 | 9 | 9 | 9 | 5 | 5 |
| Number of movies per hole | 2 | 3 | 2 | 3 | 3 | 2 |
| Number of fractions | 20 | 47 | 40 | 40 | 40 | 20 |
| Number of movies | 3541 | 2473 | 1496 | 2425 | 4339 | 1226 |
| Defocus range | 700–2200 nm | 500–2100 nm | 500–1300 nm | 500–1600 nm | 600–1900 nm | 500–2000 nm |
| Acquisition software | EPU | | | | | |
| Frame alignment and dose weighting | MotionCor2 | | | | | |
| CTF estimation | CtfFind4.1 | | | | | |
| Processing software | RELION3 | | | | | |
| Number of selected capsids | 91,445 | 89,241 | 74,760 | 129,522 | 161,596 | 35,820 |
| Capsids in final map | 41,117 | 20,251 | 44,498 | 24,854 | 44,374 | 23,384 |
| Resolution capsid | 6.1 Å | 3.7 Å | 3.0 Å | 4.2 Å | 4.0 Å | 6.0 Å |
| EMDB | | 10800 | 10803 | | 10801 | 10802 |
| extracted asymmetric units | 2,428,920 | 1,215,060 | 2,669,880 | 1,210,560 | 2,662,440 | n/a |
| Extracted units in final map | 561,731 | 919,612 | 516,257 | n/a | 383,200 | n/a |
| Resolution asymmetric unit | 5.4 Å | 3.0 Å | 3.0 Å | n/a | 3.9 Å | n/a |
| Model building, refinement and validation | - | Coot, Phenix, Chimera | Coot, Phenix, Chimera | - | - | - |

α4 of the same subunit and in contact with α3 of the other subunit (*Figure 2e*). As independently confirmed below, the packing of the extension domain to the flank of the spike was stabilized by a salt bridge between R124 in α4 of the core spike and E109 in αe2 of the extension domain (*Figure 2—figure supplement 1*).

Analysis with Intersurf (*Ray et al., 2005*) showed that the extension domain increased the inter-dimer surface area from ~1,900 Å² to ~3,400 Å², highlighting its important contribution to the intra-dimer contact. Nonetheless, the absence of folded extension domains from many spikes in freshly prepared CLPs implied that its contribution is not strictly required for capsid formation and stability, as directly addressed below.

In HBc which lacks an extension domain a disulphide-bridge between Cys61 on opposite monomers (*Figure 1—figure supplement 1*) can add to dimer stabilization (*Nassal et al., 1992*). DHBc lacks a homologous Cys residue; instead at a similar position in the spike Phe60 made a π-stacking interaction with Phe60 on the other monomer, and so did His52 with His52 (*Figure 2—figure supplement 2*). Another difference between HBc and DHBc was the position of the N-termini. In HBc the N-terminus is located at the dimer interface at the base of the spikes and contributes to the intra-dimer contact (*Wynne et al., 1999*). In DHBc the seven N-terminal residues pointed towards the lumen of the capsid, without contributing to the intra-dimer contact (*Figure 2c*). Interestingly, the two N-termini within one dimer had different orientations, with those of chains B and C directed towards the base of the spike and those of chains A and D pointing away (*Figure 2*).

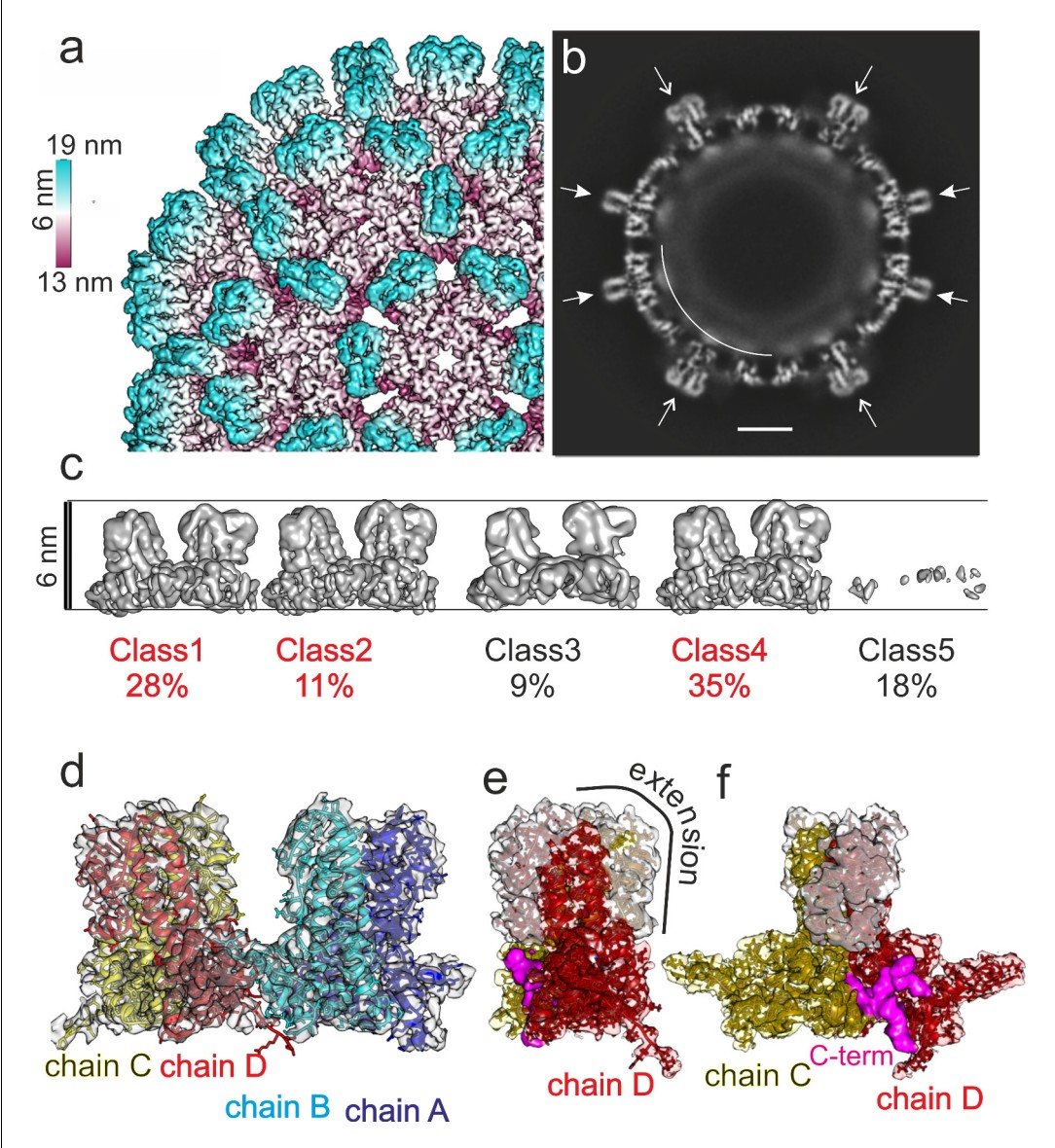

**Figure 1.** Image reconstruction of CLPs vitrified 10 months after purification. (a) A segment of the capsid color-coded by the radial position. The length of the color key corresponds to 6 nm. For comparison, the capsid of human HBc is shown in *Figure 1—figure supplement 1*. A representative micrograph and class averages are shown in *Figure 1—figure supplement 2*. The same sample 2 weeks after purification was less well resolved (*Figure 1—figure supplement 3*). Conditions for imaging and image processing are summarized in *Table 1*. (b) An equatorial slice through the EM-density with spikes indicated by arrows (narrow sections: filled arrowheads; wide cross sections: open arrowheads). The diffuse shell of density at the capsid interior is highlighted by a white arch. The scale bar indicates 10 nm. (c) Surface representations of 3D-class averages from the classification of the asymmetric units without alignment. The two horizontal lines indicate the same radial position and are 6 nm apart. Class 1, 2 and 4 differ in their radial positions by up to 2 Å. Class five represents holes in the CLPs. Perpendicular views are shown in *Figure 1—figure supplement 4*. (d) Model of the asymmetric unit of DHBc fitted into the EM-map. Chain A is shown in blue, chain B in cyan, chain C in yellow and chain D in red. The model validation is summarized in *Table 2*. The model also fits with the map of DHBc co-expressed with FkpA two weeks after purification (*Figure 1—figure supplement 5*) (e,f) Perpendicular views of the map of the CD-dimer with the model of the CD-dimer fitted. Density accounted for by the extension domain (79-119) is colored white, the core of chain C in yellow and that of chain D in red. The density at the foot of the spikes (magenta) is neither accounted by the assembly domain of chain C nor of chain D and is modelled with the most C-terminal residues.

The online version of this article includes the following figure supplement(s) for figure 1:

**Figure supplement 1.** Organization of HBc.

**Figure supplement 2.** Sample preparation and 2D-image analysis of DHBc 2 weeks and 10 months after purification.

**Figure supplement 3.** Reconstruction of CLPs vitrified 2 weeks after purification.

**Figure supplement 4.** Resolution of DHBc CLPs.

*Figure 1 continued on next page*

*Figure 1 continued*

**Figure supplement 5.** DHBc CLPs with co-expressed FkpA.

The pockets at the base of the spikes close to the N-termini of chains A and D contained density that was not accounted for by the so far described model (*Figure 1e,f* magenta). This density resembled a short peptide and was modelled with the 13 very C-terminal residues 250–262. In the other two pockets similar albeit weaker density accounted for the 6 C-terminal residues (257-262). As the connecting residues to the assembly domain were unresolved the C-termini could not be assigned to a specific subunit. Overall, the C-termini appear to take a similar structural role in DHBc as the N-termini do in HBc (*Figure 3*).

While the intra-dimer contacts differed significantly between HBc and DHBc, the inter-dimer packing in the C-terminal hand-region was largely conserved. The last resolved residues in our model agreed well with the biochemically derived C-terminal border of the DHBc assembly domain around position 187 (*Nassal et al., 2007*). At the given resolution, the fold of the hand-region was similar to that of HBc (*Figure 2d*) as it maintained the length of the α5 helix and the distance to the adjacent extended stretch. Overall, the RMSD between HBc and DHBc of the hand-region and the part of the spike below the kink and the fulcrum is 1.9 Å, further highlighting the preservation of the whole shell-region of the capsids (*Figure 3—figure supplement 1*). Consistent with the preserved inter-dimer contacts, the radius of inner and outer surface of the protein shell and the distance to a diffuse density shell in the interior of the capsid that is attributed to RNA and unresolved parts of the CTDs (*Figure 1b*) was the same in DHBc and HBc.

## Slow folding of the DHBc extension domain likely correlates with X-Pro bond isomerization

The unusually high proline content in the extension domain (*Figure 2g*, *Figure 1—figure supplement 1c,d*) suggested a cis-trans isomerization of one or more of the respective X-Pro bonds as a potentially rate-limiting step in folding of the extension domain, as it is in various proteins (*Dunyak and Gestwicki, 2016*). To test this hypothesis we co-expressed DHBc with a cytoplasmic version of the peptidyl-prolyl cis-trans isomerase FkpA, which catalyses cis-trans isomerization at prolines in *E. coli*. After 2 weeks of storage in sucrose at 4°C the respective CLPs were prepared for cryo-EM. These CLPs displayed similar overall heterogeneity as CLPs without co-expressed FkpA (*Figure 1—figure supplement 2*). However, image reconstruction revealed many CLPs with folded extension domains and better resolution (4 Å resolution, *Figure 1—figure supplement 5*, *Table 1*) than seen with the short-term stored CLPs expressed without FkpA (6.1 Å resolution). Analysis of the

**Table 2.** Validation of models.

|  | DHBc | DHBcR124EΔ |
|---|---|---|
| Non-hydrogen atoms | 6485 | 5122 |
| Modelled residues | 790 | 633 |
| Average B factors (Å²) model | 63 | 121 |
| FSC$_{0.5}$ Map$_{masked}$ vs Model | 3.9 Å | 3.1 Å |
| MolProbity score | 1.44 | 1.24 |
| Clashscore | 7 | 5 |
| Rotamer outliers (%) | 0 | 0 |
| RMSD Bond lengths (Å) | 0.003 | 0.0028 |
| RMSD Bond angles (°) | 0.41 | 0.42 |
| Ramachandran Outliers (%) | 0 | 0 |
| Ramachandran Allowed (%) | 4 | 4 |
| Ramachandran Favored (%) | 96 | 96 |
| PDB | 6ygh | 6ygi |

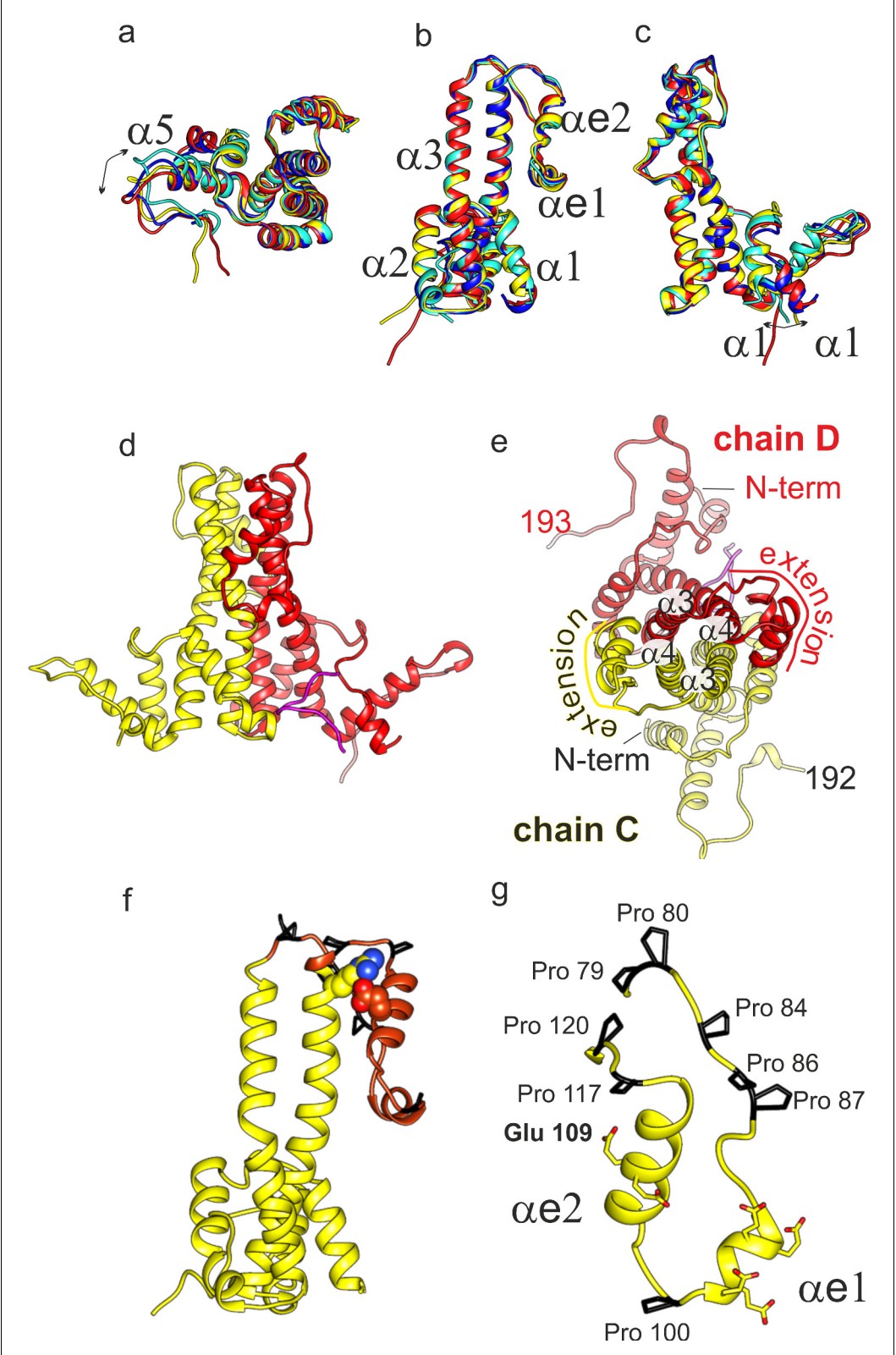

**Figure 2.** Structural model of DHBc. (**a-c**) Superposition of the four subunits in the asymmetric unit (chain A: blue, chain B: cyan, chain C: yellow, chain D red). The helices in the extension domain are labelled αe1 and αe2. All four subunits show the same overall fold with deviations in the hand-region (**a**); arrow) and at the N-terminus (**c**); arrow). (**d**) View of the CD-dimer perpendicular to the dimer axis, with the CD dimer. The C-terminus (250-262) is shown in magenta. The intra-dimer contact is stabilized by hydrophobic interactions between F60 and H52 in opposing CPs (*Figure 2—figure*

*Figure 2 continued on next page*

*Figure 2 continued*

*supplement 2*) (**e**) View of the CD-dimer along the dimer axis with the last resolved C-terminal residues of the dimer, the position of the N-termini and the position of the extension domain indicated. (**f**) Chain C with the extension domain in brown and the prolines in the extension domain in black. R124 and E109, forming the stabilizing salt bridge in the extension domain, are shown in ball representation (*Figure 2—figure supplement 1* for a close-up of the salt-bridge together with the EM-map). (**g**) Extension domain of chain C (78-121). The eight prolines are highlighted in black and the side chains of the six glutamates are indicated.

The online version of this article includes the following figure supplement(s) for figure 2:

**Figure supplement 1.** Stabilizing Elements of Extension domain.

**Figure supplement 2.** Intra-dimer contact (**a**) Close-up of the center of the CD-spike perpendicular to the dimer axis and (**b**) along the dimer axis.

asymmetric unit did not yield any evidence for unfolded extension domains (*Figure 1—figure supplement 5*). These observations support our hypothesis that cis-trans isomerization of proline is at least one of the rate-limiting steps in the structural maturation of DHBc CLPs.

## Order versus disorder in the DHBc spike tip is controlled by electrostatic interactions

The R124E mutation, like a few others, had previously been shown to increase CLP homogeneity in terms of sedimentation profile and NAGE mobility (*Nassal et al., 2007*) compared to wt DHBc CLPs; these biochemical phenotypes were referred to as class1 for wt DHBc and class2 for R124E. As the mutation reverses the sidechain charge from positive (R124) to negative (E124) an electrostatic interaction with a nearby acidic sidechain appeared plausible. Based on our structural model, R124E resides shortly after the start (P120) of helix α4 (*Figure 2*). The closest residues in space are E109 and E110 in the extension domain. When additional mutants at these positions were assessed for sedimentation and NAGE mobility (not shown) variant E109R produced a sharp R124E-like band while E110R, like R124K, showed a more WT-like fuzzy appearance. Furthermore, the double mutant E110R_R124E maintained the class 2 R124E phenotype but combining the two class2 mutations R124E and E109R in one protein (E109R_R124E) switched the profile to class1. No such switch was seen when R124E was combined with an unrelated class2 mutant, i95, which bears a five aa insertion

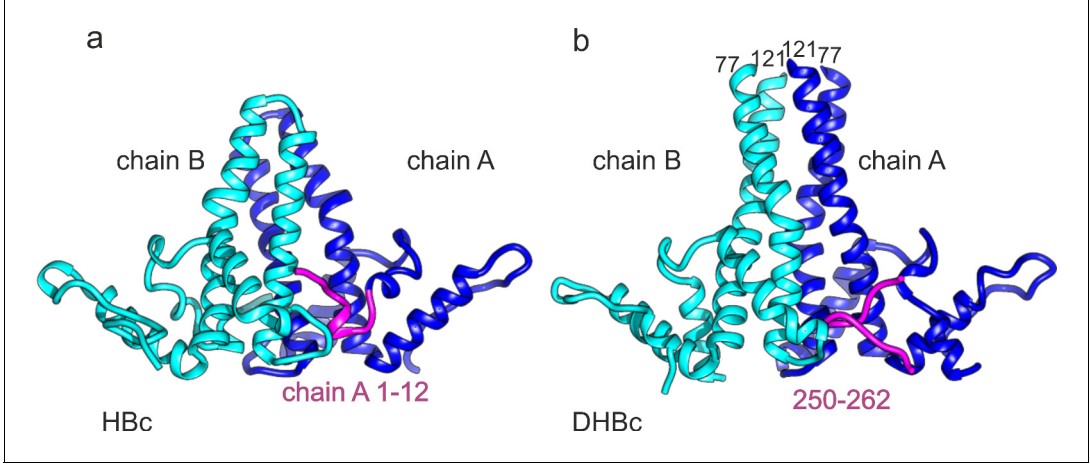

**Figure 3.** C-terminus of DHBc CPs and the N-terminus of HBc occupy the same site. (**a**) The AB-dimers of HBc (pdb: 6HTX *Böttcher and Nassal, 2018*) and b of DHBc in the same orientation. For clarity the extension domain (78-120) is not shown. Large-type and small-type CPs show a structurally conserved hand region and lower spike arrangement with an RMSD of 1.9 Å (*Figure 3—figure supplement 1*). Major differences are the packing of the four helices in the upper part of the spike and the site at the base of the spikes. This is occupied either by the 12 N-terminal residues of chain A of HBc or by the 12 C-terminal residues of DHBc (magenta). The protruding tip of the extension domain in one CP of DHBc provides a similar structural environment as the tips of the spikes in a CP dimer of HBc (*Figure 3—figure supplement 2*).

The online version of this article includes the following figure supplement(s) for figure 3:

**Figure supplement 1.** The fold of the lower part of HBc and DHBc CPs have an RMSD of 1.9 Å.

**Figure supplement 2.** The extension domain of DHBc provides a similar structural environment as the tips of the spikes in HBc.

at position 95. Together these data indicate that a salt-bridge between R124 in the spike core helix α4 and E109 in the extension domain helix eα2 is crucial for the WT-DHBc CLP phenotype. They also suggested that the more distinct behavior of the class2 CLPs relates to an inability of the mutated extension domains to properly fold.

## The DHBc extension domain is dispensable for recombinant CLP assembly

The low folding propensity of the extension domain questioned a direct role in capsid assembly. We therefore replaced residues 78–122 by a short $G_2SG_2$ linker and evaluated the consequences on assembly and particle morphology. The corresponding protein, DHBc_Δ78–122, was virtually insoluble; however, solubility was drastically improved by the additional mutation R124E (DHBcR124EΔ). Sedimentation, NAGE and negative staining EM data all indicated the formation of orderly CLPs, prompting a closer look by cryo-EM image (*Figure 1—figure supplement 2*) reconstruction. The resulting density maps at 3 Å resolution (*Figure 4*, *Figure 4—figure supplement 1*, *Figure 1—figure supplements 4* and *5*) revealed the same architecture as in the wt DHBc particles, except that all spikes remained narrow as also confirmed by the asymmetric analysis (*Figure 4—figure supplement 1*). Also, the fold of the subunits in the asymmetric unit was nearly superimposable on that of the full length protein subunits with an RMSD of 0.6 Å, except for the missing sideward pointing density. This missing density was accounted for by the 45 amino acids of the deleted extension domain whereas the flexible $G_2SG_2$ linker was not resolved. Difference mapping confirmed that all density differences between full length and internal deletion variant localized to the extension domain which therefore behaves much like an independent internal appendix.

## Folding competence of the DHBc extension domain is critical for capsid stability in eukaryotic cells and hence viral replication

To reveal a potential virological relevance of the extension domain we examined the impact of extension domain-affecting mutations on capsid expression, viral nucleic acid packaging and reverse transcription in transfected LMH cells, an avian hepatoma line supporting formation of infectious DHBV (*Condreay et al., 1990*; *Dallmeier et al., 2008*). Most variants, including the additional mutant R124Q, were tested by *trans*-complementation of a cloned CP-deficient DHBV16 genome (pCD16_core⁻) with the respective DHBc protein expressed from a separate plasmid vector. This excludes inadvertent effects on the virus genome. Mutant E109R_R124E was also tested in the *cis*-context. Controls included the non-complemented core⁻ genome and a variant with a mutated YMDD motif in the polymerase active site (pCD16_YMhD), allowing pgRNA encapsidation but not reverse transcription (*Radziwill et al., 1990*). By NAGE assay of cytoplasmic lysates and subsequent anti-DHBc immunoblotting (*Figure 5a*) the pCD16 constructs wt, YMhD and E109R_R124E, but not the core-deficient vector, produced similarly intense capsid signals whose mobilities matched those of the recombinant CLPs (lanes 16–19). Comparable signals were generated in the *trans*-format yet only with wt DHBc and mutants R124K, E109R_R124E and E110R. Variants R124Q, E109R, R124E and i95 gave very weak or no signals at all. Intriguingly, all these variants had exhibited the more homogeneous class2 phenotype when expressed in bacteria. Of the extension-less variants only Δ78–122_R124E produced specific though weak signals (*Figure 5—figure supplement 1*). To get more insight into the low or lacking capsid signals we repeated the pCD16_core⁻ cotransfections with variants R124E, Δ78–122_R124E, and E109R_R124E (including at a five-fold reduced plasmid amount) versus wt-DHBc, which reproduced the previously seen NAGE blot data for capsids and capsid-borne DNA (*Figure 5—figure supplement 2a,b*). This time, however, we additionally addressed non-assembled CP subunits, employing immunoprecipitation (IP) with a polyclonal anti-DHBc antiserum (12/99, *Vorreiter et al., 2007*) to enrich DHBc-related proteins from lysates of the transfected cells, followed by SDS-PAGE separation of the immunopellets and immunoblotting, again with the polyclonal antiserum; lysate from cells transfected with a GFP expression vector served as control. All DHBc transfected samples produced specific signals with the same mobility as the recombinant full-length proteins (*Figure 5—figure supplement 2c*); the Δ78–122_R124E band was weak, and thus consistent with the weak NAGE blot capsid signal. Strikingly, however, the R124E sample presented with a whole ladder of smaller products which in sum by far exceeded the

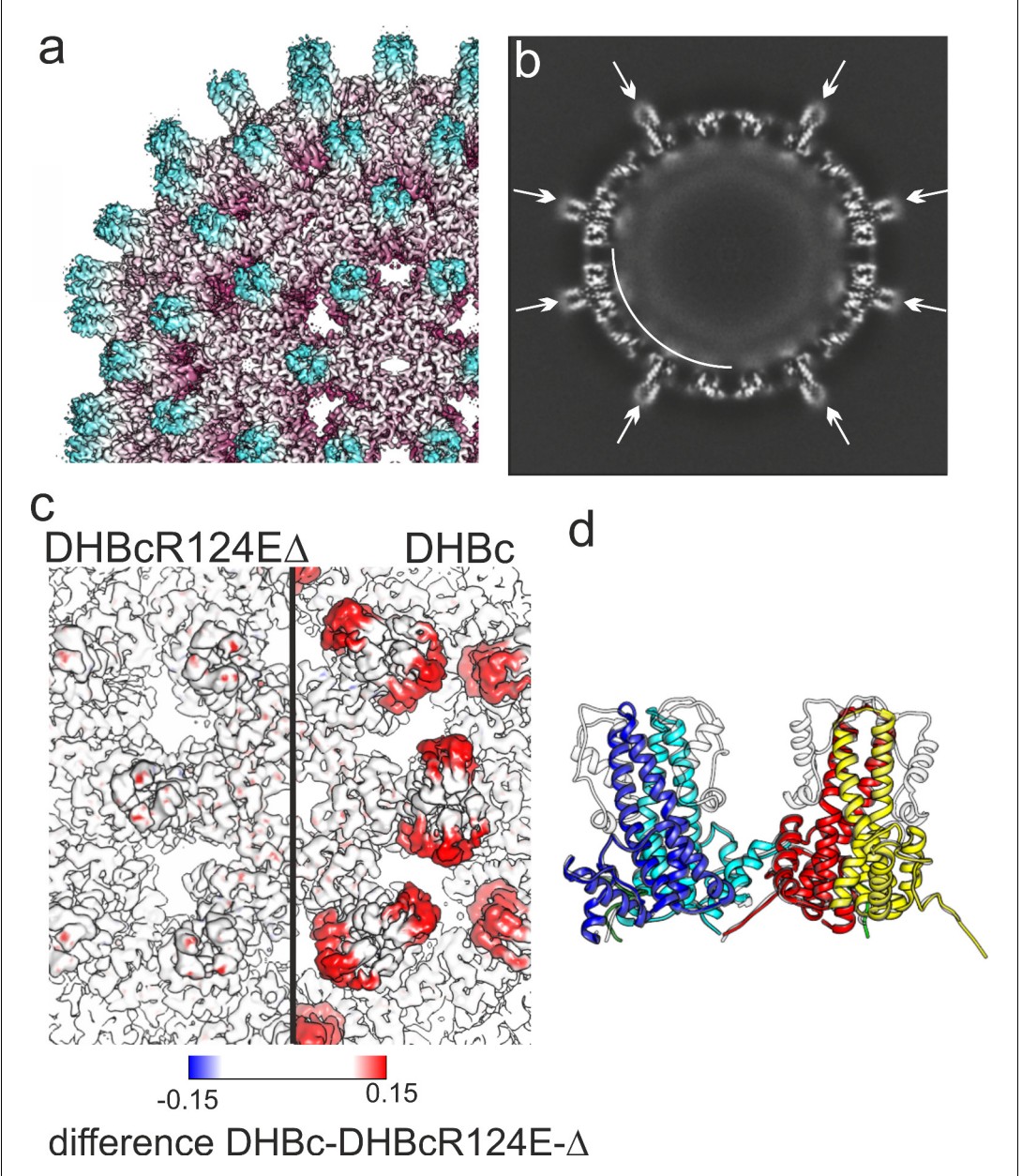

**Figure 4.** 3D-map of extension-less variant DHBc_Δ78–122 R124E (DHBcR124EΔ) in comparison to DHBc. (a) Segment of the DHBcR124EΔ capsid colored by the radial position. The Fourier-Shell correlation and local resolution analysis is shown in *Figure 4—figure supplement 1*. (b) Equatorial slice through the EM-density of DHBcR124EΔ. All spikes show narrow cross sections (arrows). A shell of diffuse density underneath the protein shell is highlighted by a white arch. (c) Close-up along the two-fold symmetry axis of a surface representation of DHBcR124EΔ (left) and DHBc (right) color-coded with the differences between the two maps. The color key is given below. Positive densities (additional density in DHBc) are shown in red and negative densities in blue (additional density in DHBcR124EΔ. (d) Model of DHBc (grey) and DHBcR124EΔ in color: chain A blue, chain B cyan, chain C yellow, chain D red are superposed). The common cores of DHBc and DHBcR124EΔ are virtually indistinguishable with an overall RMSD of 0.6 Å. The $G_2SG_2$ linker that replaces the extension domain in DHBcR124EΔ is unresolved.

The online version of this article includes the following figure supplement(s) for figure 4:

**Figure supplement 1.** Structural characterization of DHBcR124E and DHBcR124EΔ.

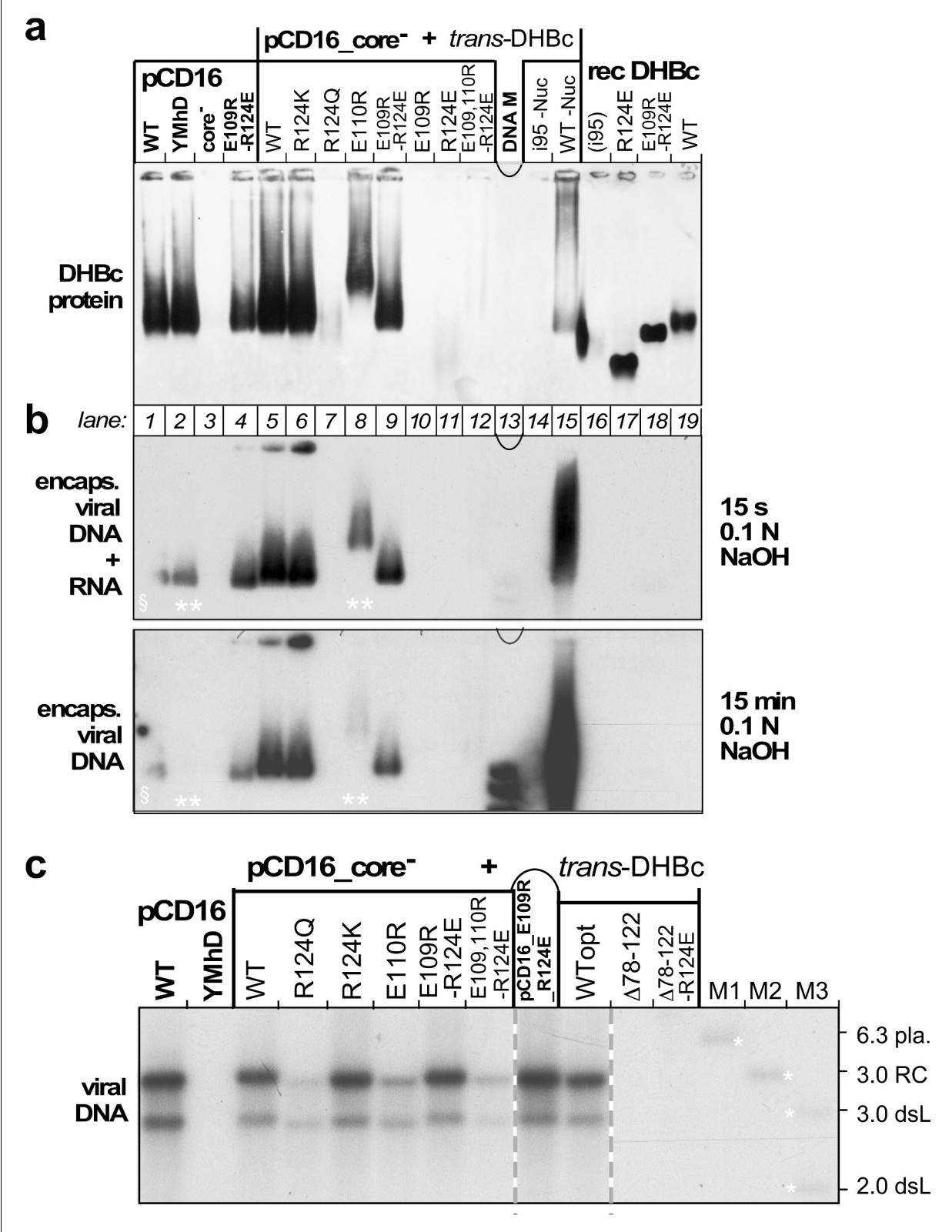

**Figure 5.** Opposite charges at DHBc positions 109 and 124 warrant stable capsid formation in hepatoma cells. (a) Capsid formation. Cytoplasmic lysates from LMH cells transfected with the indicated constructs (pCD16: mutations present in cis; pCD16_core⁻ + *trans*-DHBc: core-deficient DHBV genome co-transfected with DHBc vectors) were analyzed for capsid formation by NAGE and anti-DHBc immunoblot; recDHBc, *E. coli* derived CLPs. All samples except i95 -Nuc and WT -Nuc (lanes 14, 15) were treated with micrococcal nuclease to destroy non-packaged nucleic acids prior to NAGE. (b)
*Figure 5 continued on next page*

*Figure 5 continued*

Viral nucleic acid encapsidation. Cytoplasmic lysates separated as in (a) were examined by hybridization with a [32]P-labeled DHBV probe after opening the blotted capsids via alkali treatment for 15 s (leaving RNA and DNA intact) and, thereafter, for 15 min (hydrolyzing RNA but not DNA). Signals from pCD16_YMhD (encoding a reverse transcription-deficient polymerase) and *trans*-DHBc E110R (lanes 2, 10; marked by **) showed strong and moderate, respectively, reductions upon longer treatment. [§] in lane one indicates poor transfer of the left part of the main band. (c) Absence of grossly aberrant reverse transcription products. DNAs from cytoplasmic capsids were analyzed by Southern blotting using the same probe as in (b). Labels WT vs. WTopt refer to the DHBc nucleotide sequence from DHBV16 vs. that from the *E. coli* optimized vectors. The presence of signals for DHBc_R124Q and E109,110R_R124E likely relates to the larger input of capsids for Southern blotting than for NAGE. M1-M3, marker DNAs; pla., pCD16 plasmid. The complete blot is shown in *Figure 5—figure supplement 1*.

The online version of this article includes the following figure supplement(s) for figure 5:

**Figure supplement 1.** Extension-less DHBc variants fail to support stable nucleocapsid formation in hepatoma cells.

**Figure supplement 2.** Lack of stable capsid formation by variant DHBc_R124E in hepatoma cells correlates with increased proteolytic sensitivity.

left-over full-length band. Hence, the 124E mutation markedly enhances proteolytic susceptibility of the variant and concomitantly impedes its ability to form stable capsids in avian hepatoma cells.

To reveal capsid-borne viral nucleic acids a parallel NAGE blot was hybridized with a [32]P-labeled DHBV DNA probe, after breaking up the blotted capsids by brief exposure to 0.1 N NaOH; this leaves both RNA and DNA intact. All capsid-positive samples, except the recombinant non-viral RNA containing CLPs, scored positive in this assay (*Figure 5b*, top). Longer NaOH treatment ablated the signals for sample pCD16_YMhD and reduced it for DHBc_E110R (*Figure 5b*, bottom), in line with an exclusive and partial RNA content, respectively. Persistence of the other signals indicated they were DNA. A more sensitive Southern blot assay revealed in all these variants a wt-typical pattern of rc-DNA and dsL-DNA (*Figure 5c*). Such signals were also faintly visible for R124Q and the triple variant E109,110R_R124E but not for variant Δ78–122_R124E. In sum, only variant R124K preserving the positive charge, and the charge-polarity reversing variant E109R_R124E maintained an in all aspects wt-like replication phenotype (*Figure 5c* and *Figure 5—figure supplement 1*). Hence opposite charges at positions 109 and 124 and the extension domain´s ability to fold are critical for virus replication.

## Discussion

Despite DHBV's long history as hepadnaviral model virus, knowledge of its structural biology has lagged behind that of HBV. Our EM-map of the DHBc CLP at 3.7 Å and of the DHBc_Δ78–122_R124E CLP at 3 Å resolution allowed us for the first time to build de novo models of DHBc's assembly domain. Confirming selective aspects of evolutionary conservation between avi- and ortho-hepadnaviruses (*Kenney et al., 1995*), the new data also reveal unique structural modules, including a direct contribution of the very C-terminal residues to the capsid shell and a centrally inserted extension of ~45 aa that behaves like an independent folding entity.

### Conserved and unique structural features in DHBc versus HBc capsids

In line with earlier data (*Kenney et al., 1995*; *Nassal et al., 2007*; *Vorreiter et al., 2007*) the major type of DHBc CLP conformed to the clustered-dimer T = 4 architecture known from HBc. The last resolvable contiguous chain residues around position 190 match the biochemically derived end of the assembly domain (*Nassal et al., 2007*). Without the extension domain this length corresponds well with the ≥140 aa of the HBc assembly domain, and so does its basic fold comprising five α-helices. As in HBc, α3 and α4 constitute major parts of the intra-dimer interface with another large contribution from the folded extension domain. α3 and α4 are slightly longer in DHBc than in HBc and they differ in tilt and twist. Therefore the structure of the CPs diverges in the protruding spikes while the basic architecture is similar in the actual capsid shell including the inter-dimer contacts, mainly provided by helix α5 and the downstream sequence that folds back on it. Notably, this helical framework forms independently of the extension domain (*Figure 4*).

In HBc the N-terminal residues 1–5 embrace the base of the half-spike formed by the partner subunit in the dimer and contribute to the intra-dimer interface. In DHBc the respective positions are occupied by the C-terminal residues 250–262 while the N-termini point towards the particle lumen (*Figure 3*, *Figure 3—figure supplement 1*). This places the C-termini at a strategic position to

adopt a capsid-external disposition, e.g. for exposure of nuclear localization signals, and antibodies against the C-terminal residues of DHBc can precipitate entire capsids (*Schlicht et al., 1989*). However, our analysis of the asymmetric units in DHBc capsids did not show evidence of individual C-termini protruding further to the outside. A similar, albeit less exposed position of the C-termini underneath the base of the spikes has been observed for HBc, yet also without direct structural evidence of further externalization (*Böttcher and Nassal, 2018*). A structural role of the DHBc C-terminus is further supported by the variety of particle morphologies that result from C-terminal truncations (*Nassal et al., 2007*). Whether the DHBc sequence from positions 190 to 250 has a specific morphogenic role remains to be determined; plausibly it contributes together with packaged RNA to the ring-like capsid-internal density (*Figure 1*, *Figure 1—figure supplements 3* and *4*, *Figure 4*, *Figure 4—figure supplement 1*).

## A temporarily mobile spike extension - defining feature of hepadnaviruses employing large type CPs?

The most distinctive feature in DHBc vs. HBc are the large extensions at the tips of the spikes that comprise residues 79–119, in line with the biochemical prediction (*Nassal et al., 2007*) and directly supported by the structure of the extension-less Δ78–122_R124E CLPs (*Figure 4*). The extension sequence is unusually rich in P (8 of 45 aa, *Figure 2*) and E residues (6 of 45 aa), the two most disorder-promoting amino acids (*Theillet et al., 2013*; *Uversky, 2013*). Notably, P79/P80 and P117/P120 at the borders of the extension domain are ideally located to insulate it from the spike-forming four-helix bundle (*Figure 2*). These features are preserved in the CPs of diverse other avian hosts (*Figure 1—figure supplement 1*). Non-avian hosts with large-type CPs such as the 266 aa sequence of the Tibetan frog HBV CP diverge at many positions. Yet, the sequence aligning to the DHBc extension domain also has a very high proline content (10 of 48 aa; *Figure 1—figure supplement 1*), indicating evolutionary conservation of this feature over millions of years. We therefore expect that the fundamental structural aspects of the DHBV capsid revealed in this study are prototypical for all hepadnaviruses employing the large-type CPs. The plasticity of the extension domain is also reflected by local resolution analysis and high local B-factors in the model (*Figure 1—figure supplement 4*). Also the spike tips in HBc have high local B-factors (*Böttcher and Nassal, 2018*), perhaps in both cases facilitating induced-fit interactions, e.g. with the envelope proteins, which for DHBV might be more akin to a folding-by-binding process (*Weng and Wang, 2020*). Notably, the structure of one HBc spike tip fits well into the DHBc extension domain emerging from the core-spike which provides two such sites per spike (*Figure 3—figure supplement 2*). Thus, large-type CPs could have evolved two surface protein binding sites per spike in the extension domains, compared to only one in small-type CPs. This coincides with a generally smaller membrane part of the surface proteins in *hepadnaviridae* with large-type CPs allowing for more surface proteins per area of envelope.

Despite its disposition for disorder, the extension domain in wt DHBc can adopt a stably folded state, albeit very slowly in vitro. The time-scale in live cells is unknown but in sum our data favour that assembly of the capsid shell precedes folding of the extension domains. Assembly can occur in the complete absence of the extension domain, and CLPs with permanently disordered extension domains, such as R124E, present with more homogeneous morphologies than those from wt DHBc. Hence the large contact area to the intra-dimer interface that forms upon folding of the extension domain may even disfavour regular icosahedral assembly; conversely, the requirement for a folding helper would increase the time-span for undisturbed assembly. Thereafter, extension domain folding provides extra stability to the assembled structure. As a resolution of around 7 Å is sufficient to distinguish folded from unfolded extension domains, defining their states in various stages of the viral life-cycle, including in enveloped virions, appears worthwhile and feasible.

Based on the high density of prolines in the extension domain we expect the folding helper to be a peptidyl-prolyl cis-trans isomerase, as is supported by the higher frequency of DHBc CLPs with folded extension domains upon coexpression of *E. coli* FkpA. From such activity of a bacterial enzyme it appears unlikely that a specialized isomerase is required in DHBV host cells.

In contrast to wt DHBc CLPs, the extension domains in R124E CLPs remained invisible (*Figure 4—figure supplement 1*), suggesting the homogeneous appearance of class2 CLPs relates to mostly disordered extension domains, as supported by their similarity to the extension-less variant (*Figure 4*, *Figure 4—figure supplement 1*).

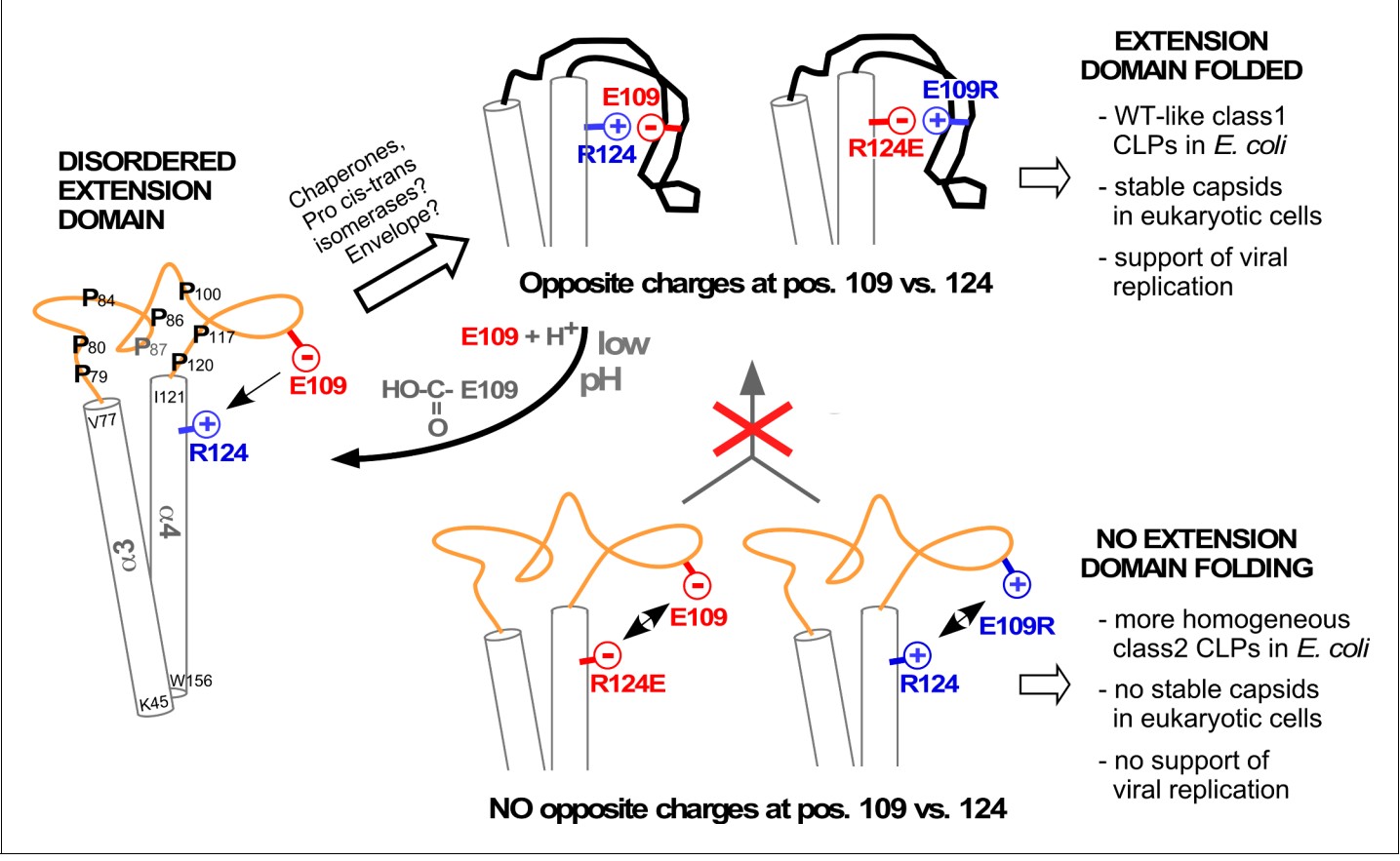

**Figure 6.** Model correlating disorder vs. order of the DHBc extension domain with phenotypes in bacteria and hepatoma cells. Opposite charges in WT-DHBc and the double mutant E109R_R124E facilitate folding of the extension domain, in cells probably accelerated by chaperones, proline cis-trans-isomerases, and/or perhaps the envelope proteins. The shape of the folded extension domain is a schematic approximation to the structural data. Protonation of the E109 carboxylate at low pH ablates the salt-bridge, possibly creating a trigger for a conformational switch. In the absence of opposite charges at positions 109 and 124 (R124E, E109R) repulsion or lack of attraction (R124Q, not shown) hamper folding of the extension domain. This may expose eukaryotic instability determinants and cause the apparent capsid instability in hepatoma cells.

Intriguingly, despite their efficient expression in bacteria DHBc class2 variants were barely if at all detectable in hepatoma cells (*Figure 5*); also Δ78–122_R124E gave only faint signals in the NAGE immunoblot (*Figure 5—figure supplement 1*). As the same eukaryotic vectors led to efficient production of wt DHBc and other class1 particles, the class2 mutations probably interfered with some aspect of protein stability. In fact, decreased CP stability and lack of capsid formation have independently been reported for a fortuitously identified variant lacking H107 (*Guo et al., 2006*) which we now know to reside within the center of helix αe2. Deletion of H107 thus puts the remaining residues of the helix out of register and, specifically, ablates a hydrophobic interaction of H107 with I99 in αe1 (*Figure 2—figure supplement 1*) which contributes to the stability of the folded extension domain.Likely, long-term unfolded extension domains enhance accessibility of instability determinants, e.g. target sites for proteases, or for ubiquitylation which initiates proteasomal degradation. This view is strongly supported by the multiple proteolysis products seen with DHBc _R124E (*Figure 5—figure supplement 2*), and the simultaneous absence of stable capsids and/or nucleocapsids from this variant (*Figure 5*;.*Figure 5—figure supplements 1* and *2*). In addition, without the extra contribution of the folded extension domains to the intra-dimer interface the capsid shell may become too labile to withstand the replication process, in particular double-stranded DNA formation. While these are testable hypotheses the lack of stable capsids from these variants precluded statements on a specific role of the extension domain in DHBV replication. Most striking, however, is the complete rescue of wt DHBV-like replication by combining two per se detrimental mutations in the double mutant E109R_R124E (*Figure 5*), fully supporting our structural model as well as the

importance of a salt-bridge between these two specific positions for the ordered state of the extension domain (*Figure 6*). Conversely, protonation of the glutamic acid carboxylate, realistically achievable at pH values of certain cell compartments (*Huotari and Helenius, 2011*), would be sufficient to break this bridge and thus create a reversible pH-responsive conformational switch for capsid stabilization and capsid disintegration.

# Materials and methods

## Key resources table

| Reagent type (species) or resource | Designation | Source or reference | Identifiers | Additional information |
|---|---|---|---|---|
| Strain, strain background (*duck*) | DHBV16 | GenBank | K01834 | duck hepatitis B virus |
| Strain, strain background (*Escherichia coli*) | BL21Star (DE3) CodonPlus RIL | Thermofisher and Agilent | C601003 #230240 | BL21Star (DE3) transformed with RIL CodonPlus plasmid |
| Cell line (*Gallus gallus*) | LMH | ATCC | CRL-2117 | chicken hepatocellular carcinoma |
| Recombinant DNA reagent | pCD16 | Nassal lab *Dallmeier et al., 2008* | | CMV promoter controlled DHBV16 vector |
| Recombinant DNA reagent | pAAV-DHBc | This paper | pAAV-MCS Addgene #46954 | authentic DHBV16 C gene cloned in pAAV-MCS2 for eukaryotic expression (see Plasmid Constructs) |
| Recombinant DNA reagent | pAAV-DHBc_Opt | This paper | pAAV-MCS Addgene #46954 | *E. coli* optimized DHBV C gene cloned in pAAV-MCS2 for (see Plasmid Constructs) eukaryotic expression |
| Recombinant DNA reagent | pRSF-DHBc_Opt | This paper *Heger-Stevic et al., 2018a* | | *E. coli* optimized DHBV C gene cloned in pRSF under T7 promoter for bacterial expression; (see Plasmid Constructs) |
| Antibody | anti-DHBc (rabbit poylclonal) | *Vorreiter et al., 2007* | serum 12/99 | raised against recombinant full-length DHBc WB (1:10,000) For immuno precipitation (IP): 50 µl Antiserum 12/99 are loaded onto 500 µl Protein A Sepharose beads (Gelbett); 30 µl loaded beads are used per IP |
| Antibody | Goat IgG anti-Rabbit IgG (Fc)-HRPO | Jackson Immuno Research | Dianova Cat. # 111-035-046 | WB (1:10,000) |
| Chemical compound, drug | TransIT-LT1 | Mirus Bio | VWR Cat. # 10767–118 | IF(1:500) |
| Software, algorithm | Relion 3 | *Zivanov et al., 2018* | | |

*Continued on next page*

*Continued*

| Reagent type (species) or resource | Designation | Source or reference | Identifiers | Additional information |
|---|---|---|---|---|
| Software, algorithm | MotionCor2 | *Zheng et al., 2017* | | |
| Software, algorithm | Ctffind4 | *Rohou and Grigorieff, 2015* | | |
| Software, algorithm | Chimera | *Yang et al., 2012* | | |

## Plasmid constructs

Bacterial DHBc vectors were based on either pET28a2-DHBc (*Nassal et al., 2007*) carrying the DHBc nucleotide sequence from DHBV16 (GenBank: K01834.1), or on pRSF_T7-DHBcOpt, a derivative of vector pRSF_T7-HBcOpt (*Heger-Stevic et al., 2018b*) in which the HBc ORF was replaced by a synthetic, *E. coli* codon usage optimized DHBc sequence (DHBcOpt). Eukaryotic DHBc expression vectors were based on the cytomegalovirus immediate early (CMV-IE) enhancer/promoter regulated plasmid pAAV-MCS (Stratagene). The parental vector pCD16 contains a 1.1x DHBV16 genome, also under CMV-IE control; in pCD16-core⁻ the initiator ATG codon of the DHBc ORF was changed to ACG. Mutations were introduced via PCR using appropriate primers and the Q5 Site-directed Mutagenesis Kit or the NEBuilder HiFi DNA Assembly Master Mix (both NEB).

The coding sequence for a cytoplasmic version (*Levy et al., 2013*) of the peptidyl prolyl isomerase FkpA (GenBank ID: AAC41459.1) was obtained by PCR on genomic DNA from *E. coli* Top10 cells, using primers FkpA_Nco⁺ 5′-CACCATCACGCCATGGCTGAAGCTGCAAAACCTG and FkpA_AvrNot⁻ 5′- GACTCGAGTGCGGCCGCCTAGGGTTTTTTAGCAGAGTCTGCGGCTTTCG (ATG start codon and complement to TAG stop codon underlined) providing terminal restriction sites to replace the Tet-promoter controlled serine-arginine-rich protein kinase (SRPK1) ORF in vector pRSF_Tet-SRPK1_T7-HBc183opt (*Heger-Stevic et al., 2018b*). All plasmid constructs were verified by DNA sequencing.

## DHBc expression and CLP enrichment

Routine bacterial expression of DHBc and its variants was performed in *E. coli* BL21* Codonplus cells at 20–25°C for 16 hr in LB medium, followed by lysozyme/sonication mediated cell lysis and sedimentation of the cleared lysates in 10–60% sucrose step gradients, all as described for HBc (*Heger-Stevic et al., 2018a*). Co-expression with FkpA was conducted analogously to HBc-SRPK1 co-expression (*Heger-Stevic et al., 2018b*), using anhydrotetracyclin for Tet promoter induction.

For cryo-EM analysis of DHBc particles a modified procedure was used, intended to improve homogeneity of the samples. To this end, BL21*(DE3) cells transformed with plasmid pRSF_T7-DHBcOpt were grown in terrific broth and induced for 16 hr at 18°C. Cells were harvested and the cell pellet was re-suspended (20% w/v) in TN300+ buffer (100 mM TrisHCl, pH 7.5, 300 mM NaCl, 5 mM β-mercaptoethanol, 5 mM MgCl$_2$, 5 mM CaCl$_2$, 2 mM PMSF and 2 mM benzamidine). Using a Microfluidizer, the cells were disrupted by 2 passages at 1,500 bar. Insoluble material was removed by centrifugation at 2,000 g and 4°C for 90 min. Then, ammonium sulphate was added to the supernatant to 40% saturation at pH 7.5 at 0°C. The resulting precipitate was isolated by centrifugation, re-solubilized, and re-precipitated by adding ammonium sulphate to 30% final saturation. The new precipitate was dissolved and subjected to density-gradient centrifugation (six steps with: 10, 20, 30, 40, 50% and 60% sucrose in 50 mM TrisHCl at pH 7.5) in an SW 32 Ti rotor (Beckman Coulter) at 4°C and 125755 g for 4 hr. After centrifugation the topmost 5 ml of the gradient were discarded and the remaining volume was manually fractionated into thirty-three 1 ml fractions. Fractions were analyzed by SDS-PAGE and NAGE as described, except that nucleic acids were stained using SYBR Safe dye (1:5000 dilution; v/v).

## Preparation of samples for electron cryo-microscopy

200 µl of the peak fractions containing about 30% sucrose were mixed with 300 µl TN50 (50 mM TrisHCl at pH 7.5 for the 2 weeks old sample) or TN20 buffer (20 mM TrisHCl pH 7.5, 50 mM NaCl,

1 mM Mg/CaCl$_2$, 5 mM DTT, for the 10 months old sample) and concentrated to 100 µl at 4°C and 2,500 g in a concentrator with a molecular weight cut-off of 100 kDa. The concentrate was five-fold diluted and re-centrifuged. This procedure was repeated until the sucrose concentration dropped below 0.03%. The final protein concentrations as determined by the Bradford assay were 2.6 mg/ml for the 2 weeks stored sample and 3.2 mg/ml for the 10 months stored sample. For the vitrification, grids (400 mesh copper grids (type R 1.2/1.3. Quantifoil Micro Tools, Jena/Germany)) were rendered hydrophilic by glow discharging in air at a pressure of 29 Pa for 2 min at medium power with a Plasma Cleaner (model PDC-002. Harrick Plasma Ithaca, NY/USA). Then, 3.5 µl of DHBc solution was pipetted onto the grids and they were plunge frozen in liquid ethane with a Vitrobot mark IV (FEI-Thermo Fisher Scientific). The settings for the Vitrobot were 3 s blot time, 45 s wait time, blot force 0 at a temperature of 4°C and 100% humidity. The vitrified samples were stored in liquid nitrogen until use.

## Electron cryo-microscopy

Grids were loaded into a Titan Krios G3 with Falcon III camera (FEI-Thermo Fisher Scientific) and movies were acquired with EPU in integrating mode for DHBc and DHBc+FkpA prepared after 2 weeks and in counting mode for DHBc prepared after 10 months, DHBc+FkpA prepared after 2 weeks and DHBc R124EΔ (*Table 1*).

## Image processing

Data of the different samples were processed separately, following similar protocols. In brief: Movies were dose-weighted and motion corrected with MotionCor2 (*Zheng et al., 2017*). Dose weighted and motion corrected averages were imported into RELION 3.0 (*Zivanov et al., 2018*) for further processing. The defocus of the dose-weighted and motion corrected movie averages was determined with ctffind4 (*Rohou and Grigorieff, 2015*). Particles were selected by the Laplacian-of-Gaussian algorithm with a mask diameter of 320 Å (data of samples prepared after 2 weeks) or with cryolo using the general network (*Wagner et al., 2019*) for all other samples followed by autopicking with RELION using 6–9 selected class averages derived from 2D-classification of the picked data. Selected particles were extracted with a box size of 440 × 440 Px$^2$ using the phase_flip option. Extracted particles were grouped by 2D-classification. Particles representing distinct capsids of the same class based on the appearance of the class averages were retained. The EM-map of F97L-capsids (*Böttcher and Nassal, 2018*) was used as starting reference (initial low-pass filter 1/25 Å$^{-1}$) for auto-refinement of the 1$^{st}$ data set (2 weeks). For the other data set (10 months) the current best map of DHBc was used as starting reference. During iterative auto-refinement icosahedral-symmetry was applied. Afterwards, particles were classified into 3 classes by 3D-classification without further refinement of orientations. The most populated 3D-class was further refined with auto-refine using only local orientation searches. This was followed by refinement of the contrast transfer function (ctf). For this, particles were re-extracted with a larger box of 800 × 800 Px$^2$ without phase-flipping followed by ctf-refinement per particle and beam-tilt refinement for the whole data set. For beam-tilt refinement, we defined beam-tilt classes for each day of acquisition to compensate for a possible drift of beam-tilt over time. For DHBcR124EΔ and DHBc+FkpA we also defined separate beam-tilt classes for each acquisition position in a hole. Ctf-refinement was followed by another round of local auto-refinement. Ctf-refinement only improved resolution of the maps of DHBc after 10 months and DHBcR124EΔ for the other maps no improvement was observed. In these cases we continued with the data without ctf-refinement.

The resolution was determined during the post-processing step for masked maps after gold-standard refinement.

## Image processing of the asymmetric unit

Starting with the results of the final refinement of the capsids, particles were symmetry expanded ('relion_particle_symmetry_expand') followed by re-extraction (128 × 128 Px$^2$), phase-flipping and re-centering to the center of one asymmetric unit. A 3D-map of the re-extracted particle fragments was calculated with 'relion_reconstruct' and a mask was generated from the model of the asymmetric unit with Chimera vop zone (*Pettersen et al., 2004*; *Yang et al., 2012*) followed by 'relion_mask_create'. The 3D-map and inverted mask were used for signal-subtraction of the extracted

particles with 'relion_project'. The signal subtracted particle images were 3D-classified without further alignment using the reconstruction of the particle fragments as starting reference and the mask generated from the asymmetric unit to focus the classification. Particles of the best resolved classes without signal-subtraction were locally refined with auto-refine using the mask of the asymmetric unit for focusing the refinement. The resolution was assessed in post-processing for masked maps after gold-standard refinement or with the local resolution option of RELION.

## Eukaryotic cell culture and transfections

Chicken LMH hepatoma cells were originally obtained from ATCC (CRL-2117) and maintained as described (*Köck et al., 2010*). Species identity of the LMH cell line was verified by amplification and partial sequencing of the mitochondrial 16S rRNA gene with primers 5'-TCCAACATCGAGGTCG TAAAC and 5'-GTACCGCAAGGGAAAGATGAA', also as described (*Köck et al., 2010*). Transfections were performed using TransIT-LT1 reagent (Mirus) as described (*Dallmeier et al., 2008*; *Schmid et al., 2011*). For *trans*-complementation of core-deficient DHBV in 6-well format 4 µg pCD16-core⁻ plasmid plus 1 µg of the desired pAAV-DHBc vector per well were used; for 10 cm dish format (*Figure 5—figure supplement 2*) DNA amounts were increased to 12.5 µg plus 2.5 µg, respectively; for sample E109R_R124E lo only 0.5 µg pAAV plasmid was employed. For non-complemented controls pAAV-DHBc was replaced by the hGFP encoding vector pTRUF5 (*Zolotukhin et al., 1996*).

## Mycoplasma testing

Potential mycoplasma contamination was monitored every fifth passage using a home-made PCR test, similar to a published procedure (*Uphoff and Drexler, 2013*). Primers UniMyco(+) 5'- CGCC TGAGTAGTATGCTCGC and MycoHom(-) 5'-GCGGTGTGTACAAAACCCGA amplify an about 520 bp segment from genomic DNA coding for the 16S rRNA of various mycoplasma species. As internal control, a pUC plasmid was constructed in which a 188 bp heterologous sequence replaces a 67 bp HpaI - BstBI fragment within a cloned 700 bp 16S rRNA gene fragment from *M. yeatzii* that contains the primer binding sites; on this template the primers amplify a 638 bp fragment. Standard 50 µl PCR reactions using KAPA HiFi polymerase (Kapa Biosystems) were set up as follows: 33 µl $H_2O$; 10 µl 5x KAPA HiFi buffer with $Mg^{2+}$; 1.5 µl 10 mM dNTP; 1.5 µl of each 10 µM primer; 0.5 µl (1 U/µl) KAPA HiFi polymerase; plus $3 \times 10E3$ or 10E4 copies of the internal control template. Cycling conditions: (1) Initial denaturation, 98°C 2 min; (2) cycling denaturation, 98°C, 10 s; (3) primer annealing, 58°C, 15 s; (4) extension, 72°C, 30 s; total 34 cycles from step (4) to (2 ; 5) final extension, 72°C, 5 min. For analysis, 5 µl of the reactions were separated by electrophoresis in 1.3% agarose gels alongside a 50 bp or 100 bp DNA ladder marker. DNAs were visualized by ethidium bromide staining. Samples generating an about 500 bp amplicon in addition to the 638 bp control amplicon were considered positive; however, all cells used in this study scored negative in this assay.

## Detection of viral capsids, capsid proteins, and capsid-borne DNA

Capsid formation by wt and mutant DHBc proteins was evaluated by immunoblotting after NAGE, using the polyclonal rabbit anti-serum 12/99 which recognizes various avihepadnavirus core proteins (*Vorreiter et al., 2007*). The same antiserum was used for immunoprecipitation of DHBc proteins in lysates from transfected cells and their detection by SDS-PAGE followed by immunoblotting on polyvinylidenfluoride (PVDF) membranes, as detailed in the legend to *Figure 5—figure supplement 2*. For detection of viral DNA in capsids NAGE gels were blotted onto Nylon membranes and hybridized with a ³²P-labeled DHBV DNA probe (*Dallmeier et al., 2008*; *Schmid et al., 2011*), after opening capsids by rinsing the membrane with 0.1 N NaOH for 15 s (preserving RNA and DNA) or 15 min (preserving only DNA). Labeled bands were visualized by phosphorimaging and/or autoradiography. Isolation of viral DNAs from cytoplasmic nucleocapsids and Southern blotting were conducted as described (*Dallmeier et al., 2008*; *Schmid et al., 2011*).

## Acknowledgements

We thank Tim Rasmussen, Julian Lenhart, Christian Kraft and the students of the master course electron microscopy 2019 in Würzburg for data acquisition, and Andrea Pfister for help with recombinant

CLP expression, and Ida Wingert and especially Peter Zimmermann for cell culture experiments. MN is grateful to the DFG for support during the earliest phase of this work (grant Na154/9-4) and BB during the latest phase of this work (BO1150/17-1). Cryo-EM was carried out in the cryo-EM-facility of the Julius-Maximilian University Würzburg (INST 92/903-1FUGG).

## Additional information

### Funding

| Funder | Grant reference number | Author |
|---|---|---|
| Deutsche Forschungsge-meinschaft | BO1150/17-1 | Bettina Böttcher |
| Deutsche Forschungsge-meinschaft | INST 92/903-1FUGG | Bettina Böttcher |
| Deutsche Forschungsge-meinschaft | Na154/9-4 | Michael Nassal |

The funders had no role in study design, data collection and interpretation, or the decision to submit the work for publication.

### Author contributions

Cihan Makbul, Data curation, Formal analysis, Validation, Investigation, Visualization, Methodology, Writing - original draft, Writing - review and editing; Michael Nassal, Conceptualization, Resources, Funding acquisition, Validation, Investigation, Visualization, Methodology, Writing - original draft, Project administration, Writing - review and editing; Bettina Böttcher, Conceptualization, Data curation, Formal analysis, Funding acquisition, Validation, Investigation, Visualization, Methodology, Writing - original draft, Project administration, Writing - review and editing

### Author ORCIDs

Michael Nassal (iD) https://orcid.org/0000-0003-2204-9158
Bettina Böttcher (iD) https://orcid.org/0000-0002-7962-4849

### Decision letter and Author response

Decision letter https://doi.org/10.7554/eLife.57277.sa1
Author response https://doi.org/10.7554/eLife.57277.sa2

## Additional files

### Supplementary files

• Transparent reporting form

### Data availability

EM-maps are deposited in the EMDB. Where applicable models were deposited in the pdb DHBc capsid: 10800 (EMDB) 6ygh (pdb) DHBC co expressed with FkpA: 10801 (EMDB) DHBC R124E (mutant): 10802 (EMDB) DHBCR124E_del (deletion-mutant): 10803 (EMDB) 6ygi (pdb).

The following datasets were generated:

| Author(s) | Year | Dataset title | Dataset URL | Database and Identifier |
|---|---|---|---|---|
| Makbul C, Nassal M, Bottcher B | 2020 | DHBc capsid | https://www.rcsb.org/structure/6ygh | RCSB Protein Data Bank, 6ygh |
| Makbul C, Nassal M, Bottcher B | 2020 | DHBCR124E_del (deletion-mutant) | https://www.rcsb.org/structure/6ygi | RCSB Protein Data Bank, 6ygi |
| Makbul C, Nassal M, Bottcher B | 2020 | DHBc capsid | https://www.ebi.ac.uk/pdbe/entry/emdb/EMD-10800 | Electron Microscopy Data Bank, EMD-10 800 |

| | | | | |
|---|---|---|---|---|
| Makbul C, Nassal M, Bottcher B | 2020 | DHBC co expressed with FkpA | https://www.ebi.ac.uk/pdbe/entry/emdb/EMD-10801 | Electron Microscopy Data Bank, EMD-10801 |
| Makbul C, Nassal M, Bottcher B | 2020 | DHBC R124E (mutant) | https://www.ebi.ac.uk/pdbe/entry/emdb/EMD-10802 | Electron Microscopy Data Bank, EMD-10802 |
| Makbul C, Nassal M, Bottcher B | 2020 | DHBCR124E_del (deletion-mutant) | https://www.ebi.ac.uk/pdbe/entry/emdb/EMD-10803 | Electron Microscopy Data Bank, EMD-10803 |

The following previously published datasets were used:

| Author(s) | Year | Dataset title | Dataset URL | Database and Identifier |
|---|---|---|---|---|
| Mandart E, Kay A, Galibert F | 1984 | Nucleotide sequence of a cloned duck hepatitis B virus genome: comparison with woodchuck and human hepatitis B virus sequences | https://www.uniprot.org/uniprot/P0C6J7 | Uniprot, P0C6J7 |
| Mandart E, Kay A, Galibert F | 1984 | Nucleotide sequence of a cloned duck hepatitis B virus genome: comparison with woodchuck and human hepatitis B virus sequences | https://www.ncbi.nlm.nih.gov/nuccore/K01834 | NCBI Nucleotide, K01834.1 |
| Horne SM, Young KD | 1995 | Escherichia coli and other species of the Enterobacteriaceae encode a protein similar to the family of Mip-like FK506-binding proteins | https://www.ncbi.nlm.nih.gov/protein/AAC41459.1 | NCBI Protein, AAC41459.1 |
| Böttcher B, Nassal M | 2018 | WT Hepatitis B core protein capsid | https://www.rcsb.org/structure/6HTX | RCSB Protein Data Bank, 6htx |
| Leslie AGW, Wynne SA, Crowther RA | 1999 | HUMAN HEPATITIS B VIRAL CAPSID (HBCAG) | https://www.rcsb.org/structure/1QGT | RCSB Protein Data Bank, 1qgt |
| Galibert F, Mandart E, Fitoussi F, Tiollais P, Charnay P | 1979 | Nucleotide sequence of the hepatitis B virus genome (subtype ayw) cloned in E. coli | https://www.uniprot.org/uniprot/P03146 | Uniprot, P03146 |
| El-Gebali S, Mistry J, Bateman A, Eddy SR, Luciani A, Potter SC, Qureshi M, Richardson LJ, Salazar GA, Smart A, Sonnhammer ELL, Hirsh L, Paladin L, Piovesan D, Tosatto SCE, Finn RD | 2019 | Hepatitis core antigen | http://pfam.xfam.org/family/PF00906 | Pfam, PF00906 |

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
