## [Decision Letter]

**Acceptance summary:**

Duck hepatitis B virus (DHBV) has been used widely as a model to study the hepatitis B virus (HBV) – a human virus infects over 250 million people worldwide. This work revealed the structure of DHBV core protein, which is substantially larger than that of HBV. The study generates important new information regarding the assembly and replication of HBV-like viruses (hepadnaviruses).

**Decision letter after peer review:**

Thank you for submitting your article "Slowly folding surface extension in the prototypic avian hepatitis B virus capsid governs stability" for consideration by *eLife*. Your article has been reviewed by two peer reviewers, and the evaluation has been overseen by a Reviewing Editor and Cynthia Wolberger as the Senior Editor. The reviewers have opted to remain anonymous.

The reviewers have discussed the reviews with one another and the Reviewing Editor has drafted this decision to help you prepare a revised submission.

We would like to draw your attention to changes in our revision policy that we have made in response to COVID-19 (https://elifesciences.org/articles/57162). Specifically, we are asking editors to accept without delay manuscripts, like yours, that they judge can stand as *eLife* papers without additional data, even if they feel that they would make the manuscript stronger. Thus the revisions requested below mainly address clarity and presentation.

Summary:

The duck HBV (DHBV) has been used widely as a model to study the human pathogen, the hepatitis B virus (HBV). A striking difference between HBV and DHBV is that the DHBV capsid protein is substantially larger than that of HBV (ca. 260 vs. ca. 180 amino acid residues). The three significant accomplishments of this paper are (i) the observation that DHBV capsid protein is partially unfolded and yet still assembles, (ii) the protein will slowly refold and that can be catalyzed by the presence of proline isomerase, and (iii) the actual structure of the protruding domain.

Essential revisions:

There are some points to be addressed or discussed to further strengthen the manuscript, for example:

It is well known that HBc lacks the extension domain and yet is fully functional in human and avian cells. Conversely, DHBc with the extra domain is also functional in both human and avian cells. If there were indeed a host-cell specific determinant for the folding/functions of the extension domain in DHBV, as implied by the authors, it would be interesting to test the various extension domain deletion/substitution mutants in human cells, in addition to avian cells. As it stands now, little could be learnt regarding the exact roles of this extension in viral replication from the phenotype analysis of these mutants in avian cells, due to the apparently severe deleterious effect on capsid assembly.

Figure 5. The authors should check levels of the mutant DHBV capsid protein subunits using SDS-PAGE and western blotting. Did the subunits still accumulate? Was the defect in subunit stability, assembly, or the stability of the assembled capsids?

The question of which chaperone is responsible for capsid protein refolding in vivo is not answered in this paper. The effect of proline isomerase indicates that refolding can be catalyzed. However the limited effect of proline isomerase suggests that some other chaperone(s) may be involved. Also, the interaction of the capsid protein with the surface protein may also contribute to folding. The possibilities should be discussed with an open mind.

The evolutionary connections from avian to other HBVs is important to understanding virus evolution. The authors should also relate the DHBV structure to other insert containing sequences instead of just the limited sequence alignment of Figure 1—figure supplement 1D.

Following please find more comments from the two reviewers.

Reviewer #1:

The duck HBV (DHBV) has been used widely as a model to study the human pathogen, the hepatitis B virus (HBV). A striking difference between HBV and DHBV is that the DHBV capsid protein is substantially larger than that of HBV (ca. 260 vs. ca. 180 amino acid residues). The authors report high-resolution structures of several DHBV capsid-like particles (CLPs) determined by electron cryo-microscopy. These structures show that DHBV CLPs are generally similar to those of HBV, consisting of a dimeric α-helical backbone with protruding spikes at the dimer interface. The ca. 45 amino acid proline-rich extension unique to the DHBV capsid forms an extra domain near the spike as compared to HBV. The authors noticed that folding of the extension domain seemed to take months in vitro, which could accelerated by prolyl peptidyl isomerase, consistent with a catalyzed process in vivo. DHBc variants lacking a folding-proficient extension domain were competent to assemble capsids in bacteria but failed to form stable nucleocapsids in avian cells. The authors propose that the extension domain acts as a conformational switch with differential response options during viral infection.

This is an interesting study that has a strong rationale and generates important new information regarding the assembly and replication of HBV-like viruses (hepadnaviruses). The authors should consider the following points in revising the manuscript.

Reviewer #2:

In this paper the authors show the structure of Duck Hepatitic B Virus Core Protein and variants. DHBc is a dimeric protein that assembles into T=4 capsids. While closely related to the human homolog, DHBc has a bulky insert at the dimer interface. The authors show:

1) The insert is disordered in freshly purified protein

2) Can slowly fold, over several months, to an ordered form

3) Co-expression in *E. coli* with a prolyl isomerase yields capsids that are somewhat better folded.

4) Compensating charged residue mutations support a stable insert. Non-compensated mutations do not. The resulting proteins with a misfolded insert do not support assembly of functional capsid.

5) Deletion of the disordered residues results DHBcR124E∆ which has a well-ordered assembly domain and shows how the helical dimer interface is preserved

The most striking feature of the results is the folded-unfolded transition of the insert as discussed by the authors. This insert is present in lizard and fish variants of HBV but not in mammalian variants. There is an evolutionary argument. The authors do not discuss whether the insert may also interact with receptors or other host-protein.

The authors include a peculiar discussion of a gap at the dimer interface which may be "pocket factor" binding site. Has a pocket factor ever been observed or is this a speculation? The authors should clarify if this discussion is entirely speculative.

The quality of the reconstructions is high. However many of the figures are too small and busy. The text is clear with a few exceptions. The structure of the insert will facilitate future experimentation and will raise interest in non-mammalian HBV viruses. The puzzle of the folded to unfolded transition is striking. Co-expression of DHBc with proline isomerase yields a marginally better reconstruction. In the model structures, are prolines in the cis conformation? Would any chaperone do? The conclusion that the delay in folding is due to proline isomerization is ambiguous.

---

## [Author Response]

Essential revisions:There are some points to be addressed or discussed to further strengthen the manuscript, for example:It is well known that HBc lacks the extension domain and yet is fully functional in human and avian cells. Conversely, DHBc with the extra domain is also functional in both human and avian cells. If there were indeed a host-cell specific determinant for the folding/functions of the extension domain in DHBV, as implied by the authors, it would be interesting to test the various extension domain deletion/substitution mutants in human cells, in addition to avian cells. As it stands now, little could be learnt regarding the exact roles of this extension in viral replication from the phenotype analysis of these mutants in avian cells, due to the apparently severe deleterious effect on capsid assembly.Figure 5. The authors should check levels of the mutant DHBV capsid protein subunits using SDS-PAGE and western blotting. Did the subunits still accumulate? Was the defect in subunit stability, assembly, or the stability of the assembled capsids?

We agree that the biological function of the extension domain in the large-type Cp remains an intriguing open question, and that investigating the impact of a different, e.g. human, cell environment is a potentially promising approach for new insight. An example is our demonstration, some time ago that the avian DHBV produces much more cccDNA in human hepatoma cells than human HBV. These phenotypic differences might indeed relate to the presence vs. absence of the extension domain in the respective Cps. Hence these and further host-factor focussed investigations are certainly worthwhile but in our view are beyond the scope of the current structure-focussed study.

However, we experimentally followed up the reviewer suggestion to address the fate of the unassembled subunits of the key mutants DHBcR124E and DHBcR124E∆ which failed to generate stable, replication-competent nucleo-capsids in chicken LMH cells – whereas DHBc WT as well as the charge-reversal mutant DHBc_E109R_R124E did. This was initially shown by DHBc capsid and DHBV DNA detection after NAGE (Figure 5). As direct SDS-PAGE immunoblotting of cell lysates did not yield conclusive data, we adopted an immunoprecipitation (IP) protocol to enrich potentially low amounts and/or partially degraded Cp subunits (about half of the lysate from a 10 cm dish as IP input). To cover as many DHBc epitopes as possible we used a polyclonal rabbit antiserum (12/99; Vorreiter et al., 2007) both for the IP and the subsequent immunoblot (IB) detection step. To minimize signals from the IP antibody chains on the blot we then used an Fc-part specific anti-rabbit secondary antibody-peroxidase conjugate in the final visualization step (which does not react strongly with the antibody light chains in the relevant 25 kDa range of the gel), producing the data now shown in Figure 5—figure supplement 2C. To corroborate the previously seen nucleocapsid phenotypes in the very same source samples as used for the IP/IB analysis we also subjected about 5% of each lysate to the NAGE-based DHBV capsid and capsid-associated DNA assays (Figure 5—figure supplement 2A, B).

As before the extension-less variant DHBcR124EΔ gave a very weak NAGE capsid signal (and no detectable capsid DNA signal) and this correlated with a weak signal in the SDS-PAGE IB, comparable to the E109R_R124E lo sample that had received one fifth the amount of DHBc plasmid DNA of the others. Hence the lower steady-state level of DHBcR124EΔ subunits (be it from less efficient production or more efficient degradation) might explain the lower levels of capsids detected in the NAGE IB; however, as previously surmised, the complete lack of detectable DHBV DNA supports a replicative defect of the mutant (pgRNA packaging, and/or reverse transcription). This assumption is now further strengthened by the detectability of viral DNA in the DHBcR124EΔ lo sample, which produced a similarly low capsid signal.

The most striking new data concern variant DHBc_R124E which, as before, gave no capsid and no capsid-DNA signal in the NAGE blots (although the *E. coli* derived faster-than-wt migrating R124E capsids were clearly detectable on the same blot). The IP/SDS-PAGE/IB protocol revealed the presence of substantial amounts of DHBc-related signals, however, besides little full-length protein the faster mobility of the other bands indicates heavy, variant-specific proteolytic processing, as similar products are completely absent from the DHBc WT and E109R_R124E samples.

One can still argue whether a lack of assembly competence of R124E in eukaryotic cells (vs. clear capsid formation in bacteria) causes higher sensitivity to proteolysis, or vice versa. However, although these new data do not answer all of the reviewer´s respective questions they directly support the previously only assumed defect in subunit stability when no salt bridge can form between positions 109 and 124 and when the extension domain does not fold.

We have added these new data as Figure 5—figure supplement 2, and added statements on the increased proteolysis susceptibility of DHBc_R124E in eukaryotic cells in the Results (subsection “Folding competence of the DHBc extension domain is critical for capsid stability in eukaryotic cells and hence viral replication”) and the Discussion section (subsection “A temporarily mobile spike extension – defining feature of hepadnaviruses employing large type CPs?”) of the main text. We also added additional information to the Materials and methods (subsection “Detection of viral capsids, capsid proteins, and capsid-borne DNA”).

The question of which chaperone is responsible for capsid protein refolding in vivo is not answered in this paper. The effect of proline isomerase indicates that refolding can be catalyzed. However the limited effect of proline isomerase suggests that some other chaperone(s) may be involved. Also, the interaction of the capsid protein with the surface protein may also contribute to folding. The possibilities should be discussed with an open mind.

Yes, it is true that we cannot say which chaperone is (or which chaperones are) acting in vivo. However, we respectfully disagree with the reviewer about the "limited effect" of the PPiase. While after two weeks and 2 months (not shown) without co-expression of a PPiase, most DHBc subunits in capsids have still unfolded extension domains this has changed to the majority of extension domains being folded within 2 weeks when co-expressed with a PPiases. A general PPiase is sufficient for the catalysis as it works with the *E. coli* FkpA PPiase. Typically, PPiases are among the most abundant proteins. Therefore, we would expect that the extension domain folds catalysed by a general PPiase of the host cells.

Another effect on the folding propensity of the extension domain is the formation of salt bridges (R124-E109) and hydrophobic interactions (H109-I99). These interactions are unlikely to require additional chaperones for formation and might happen instantaneously after the prolines are in the correct isomerization state. More importantly the significance of these stabilizing interactions can be interrogated by mutations: Mutants with a destabilized extension domain (DHBc R124E) behave very similar in cell culture to mutants with an entirely missing extension domain (DHBc R124E∆) as they do not form stable capsids that support replication. Considering that the folded extension domain almost doubles the intra-dimer interface it is conceivable that *both* the mutant capsids are less stable. In line with a reduced stability of capsids harbouring CPs with disordered extension domains, we observed that DHBcR124E∆ and DHBcR124E CLPs burst in thin, vitrified buffer whereas DHBc CLPs remains stable. Moreover, the new cell culture data (see above) illustrate a clearly enhanced proteolysis susceptibility of the R124E variant in cell culture.

Thus proline isomerization probably acts by delaying the folding of the extension domain, whereas interactions within the extension domain are likely to fine-tune capsid stability during the viral life cycle. We have broadened the Discussion along these lines (subsection “A temporarily mobile spike extension – defining feature of hepadnaviruses employing large type CPs?”)

The evolutionary connections from avian to other HBVs is important to understanding virus evolution. The authors should also relate the DHBV structure to other insert containing sequences instead of just the limited sequence alignment of Figure 1—figure supplement 1D.

While avian large-type CPs have a well preserved extension domain, the sequence conservation of the extension domain to large-type CPs in other species is less well preserved. However, a large number of prolines in the extension domains of other host species is maintained. This is what we wanted to show with the alignment. Detailed, sequence based discussions of evolutionary aspects have been made by Lauber et al., 2017, and Dill et al., 2016, which is given in the Introduction.

Based on the weak sequence conservation we would not necessarily expect that amphibian or lizard extension domains have a very similar structure – although structural data in Lauber et al., 2017, suggest that small-type CPs of nackedna viruses adopt an HBV CP-like fold despite highly divergent primary sequence. We have modelled the structure of Tibetan Frog CPs based on our DHBc structure. This suggests indeed a similar fold of the extension domain with maintenance of αe2 but not of αe1 and the positioning of the prolines at similar strategic positions as in avian viruses (Author response image 1). However, a large none-helical insert of some 20 residues with 2 prolines in α3 at the base of the core-spike raises doubts about details of the model. Thus a proper structure should be more revealing as to which extent salt bridges, charge distribution and hydrophobic interactions are maintained in the extension domains of other large-type CPs. Considering the low conservation and the lack of structure of other large-type CPs, we have chosen not to include a structure-based discussion on evolution of the large-type CPs.

**Author response image 1. sa2fig1:** Model of the large-type CP of Tibetan frog. The model was generated with Modeller (Webb and Sali Current Protocols in Bioinformatics 54, John Wiley and Sons, Inc, 5.6.1-5.6.37, 2016.), using our DHBc structure as template. Prolines are highlighted in black.

However, large-type CPs go together with small-type surface proteins. We think that the large type CPs have evolved two potential surface protein binding sites per spike compared to only one in small type CPs. Support for this idea comes from the observation that the tip of the small-type HBc CPs matches in its overall fold and arrangement of loops the upper part of one extension domain. We have added the information on the size of the surface proteins in the Introduction (second paragraph) and have discussed the aspect of two potential binding sites per spike in the Discussion (subsection “A temporarily mobile spike extension – defining feature of hepadnaviruses employing large type CPs?”). We also added a figure to illustrate the point (Figure 3—figure supplement 2).